# "Do we even have a voice?" Health providers' perspective on the patient accommodation strategies in Bangladesh

**Md. Ruhul Kabir**[1,2]*, **Kara Chan**[1]

**1** School of Communication and Film, Hong Kong Baptist University, Kowloon Tong, Hong Kong,
**2** Department of Food Technology & Nutrition Science, Noakhali Science & Technology University, Noakhali, Bangladesh

* 20481713@life.hkbu.edu.hk, ruhul109@gmail.com

**Data Availability Statement:** All relevant data are within the paper.

**Funding:** The author(s) received no specific funding for this work.

## Abstract

### Objective

In a resource-constrained setting like Bangladesh, effective patient-provider communication is critical to the delivery of maternal healthcare services. Using communication accommodation theory, this study tried to understand how providers perceive, engage, and accommodate patients' needs in maternity wards.

### Methods

This qualitative exploratory study used a semi-structured interview guide to conduct in-depth face-to-face interviews of ten healthcare providers in two government-funded public health facilities in Bangladesh. The interview data were analyzed using MAXQDA 2020 software.

### Results

The thematic analyses revealed that nurses and midwives faced conceivable neglect from patients and their attendees due to possible service and facility constraints, indicating their low status and control within the service operation. Despite efforts to address patients' emotional and psychological concerns, providers appear to avoid (divergence strategy) confronting patients and their irate visitors. Unimodal convergence emerged when providers accepted patients' arguments about the systematic inadequacy of service facilities. Providers have employed interpretability tactics to communicate medical opinions and applied nonverbal cues where necessary.

### Conclusion

A lack of open communication between healthcare providers and patients necessitated various forms of accommodation. Integrated strategies addressing service restrictions and initiatives fostering patient understanding and cooperation may improve patient-provider communication.

**Competing interests:** The authors have declared that no competing interests exist.

## Introduction

Experiencing mistreatment, discrimination, and disrespect from healthcare providers in a resource-constrained setting in low-and-middle countries are some of the reported misconducts that many studies have reported [1–3]. According to the World Health Organization (WHO), these reported misconducts during childbirth, where stakes are high, violate the patients' rights, which marks the low quality of healthcare [4]. Provision of good quality care and patients' ability to avail and access them remains a global health priority, especially for women during childbirth [5].

Health providers and patients should maintain a good, mutually respectful relationship. To build a therapeutic patient-provider relationship, effective communication acts as an important prerequisite [6]. Despite the importance of effective patient-provider communication in achieving better health outcomes, healthcare providers' perspectives on this issue remain scarce and limited, particularly where inequalities in systematic healthcare infrastructure are recognizable [7]. More importantly, the provider's perceptions and experience on the degree of interaction with pregnant women have rarely been described qualitatively, particularly in the context of the developing world.

A study conducted in Ethiopia offered insight into provider perspectives, demonstrating that providers had witnessed disrespectful behavior and improper service delivery. More than half of the respondents felt mistreated and abused on the job [8]. According to a separate study conducted in Nigeria, the majority of alleged disrespect and mistreatment was not deliberate but rather the result of poor patient compliance during childbirth [9]. A closer examination of this topic demonstrates the impact of the imbalance of power between doctors and patients. There have been reports of sexism, gendered norms, and social inequalities in obstetric care [10]. While there is evidence of alleged patient misconduct during childbirth, the perspectives of healthcare providers on these possible allegations are extremely sparse [1].

It is imperative to analyze how health care providers (especially nurses and midwives) recognize the overall relationship with patients and their attendees and how they accommodate patients in a setting with limited resources. According to a qualitative study conducted in Mozambique, midwives were disrespected by the larger community due to their inferior status compared to physicians and were regularly held accountable for poor health outcomes [11]. However, the interaction of healthcare providers with patients was not thoroughly studied, nor were possible patient accommodation measures necessary for developing a therapeutic relationship. This study tried to address a vacuum in the literature by focusing on the provider's perspective on the overall scenario of interaction in the maternity wards during childbirth, what encourages or stifles interaction, and how patients are cared for. The communication accommodation theory [12] is extrapolated to understand the health providers' accommodation techniques in response to patient needs and expectations. This study seeks to advance knowledge by examining the context of Bangladesh, where the optimal level of maternal health service usage is very low [13], despite widespread concern about maternal deaths.

### Literature review

**Importance of patient-provider communication on effective maternal healthcare service delivery.** Patient-provider communication and positive attitudes and rapport are fundamental in effective maternal service delivery operations [14]. A descriptive qualitative study conducted in Malawi found that women who had satisfactory communication with health workers were encouraged to deliver at the health facility. The inductive thematic analysis highlighted the need to devise an intervention design to improve provider-patient interactions [3]. Notably, excellent communication promotes timely management of delivery

complications. Plus, it helps clinicians work together with patients and their families to make constructive medical decisions regarding pregnancy care. A study of immigrant women in Greater London confirms these findings. The in-depth discussion with patients revealed the relevance of professionalism and general competency as superior attributes of providers when it comes to interaction [15].

**Barriers and drivers of mistreatment in receiving maternal healthcare.**   Sen et al. (2018) explored the intersection of socioeconomic disparities, institutional frameworks, and the delivery process to identify plausible drivers of abuse and disrespect in obstetric care [10]. In explaining the role of social and economic disparities in receiving maternal care, one study portrayed the deep-rooted gender discrimination in society as one of the reasons [16]. Additionally, underprivileged people subjected to long-term mistreatment become acclimated to the disrespect they receive and cannot recognize inadequate healthcare because of this normalization and tolerance. Disrespectful behavior results from physicians' low interest in treating or learning about the patient's medical history [17].

Following this, it's important to consider how the structure and processes of healthcare organizations can affect service delivery, particularly in light of resource shortages and the lack of defined standards and norms [10]. Resource constraints, exhaustion, and demanding work hours have been identified as stressors that exert behavioral influence. Inadequate supplies and equipment, language barriers, patient and family disdain, impatience, and uncooperative behaviors contribute to undesirable scenarios during facility delivery [1]. Facility culture that does not ensure accountability can also contribute to poor judgment of actions from some providers [1]. Another study in the UK focused on midwives' perceptions of ethnically based inequities in maternity wards, highlighting the significance of interagency collaboration across health and care institutions and increasing midwives' cultural competencies to deal with ethnically diverse populations [18].

**Providers-patient communication in Bangladesh perspective.**   More than 99 percent of maternal deaths occur in developing countries, with Sub-Saharan Africa and Southern Asia bearing the brunt of most of these losses [4]. Bangladesh, a South Asian developing country experiencing a very high maternal mortality rate (MMR), and deplorably the MMR stalled to around 190 deaths per 100,000 live births in the last decade. This slowed rate begs the issue of whether improving overall healthcare quality and provider-patient collaboration could result in improved health outcomes [19]. Islam et al. (2015) reiterated that the general quality of healthcare at Bangladesh's district and subdistrict level health facilities is substandard due to a lack of healthcare workers, equipment, medical supplies, and logistics. As a result of the institutional structure and procedure outlined previously, health practitioners acknowledged many limiting constraints that prevent them from providing high-quality care. Lack of staff training and laboratory assistance hinders the entire service delivery process [20]. Another study in Bangladesh also reported negative attitudes, negligence, and delay in service [21]. However, considerable research is needed in this area, and providers' narratives must be included along with patient management strategies. Their patients' accommodations strategies need to be understood and studied to better understand the situation.

**Communication accommodation theory (CAT) and patient-provider communication.**   Communication accommodation theory (CAT), one of the behavioral theories of communication [22], applies to certain situations where interactions are evident among people from different cultures and languages [12]. The theory applies a framework to perceive the interpersonal and intergroup dynamics where communicators try to adjust their verbal (e.g., language) and non-verbal (e.g., facial expression, tone, eye contact, etc.) cues to accommodate each other [23]. The degree and quality of interaction depend on the motivation and ability to adjust. CAT focuses on how and when people adjust (accommodate or non-accommodate) to

each other and the societal ramifications of those adjustments, notably in handling conflicts [23, 24]. CAT can be applied to understanding patient-provider interaction as the theory act as a functional interface between communication, linguistics, and social psychology [12]. Individuals' beliefs and motivations are the underlying features of communicative behaviors that either converge, diverge or maintain relative to one another in the immediate situation. Convergence and divergence imply adapting one's communicative behaviors and patterns to be more similar or dissimilar to others, respectively. These adjustments can take several forms based on social value (upward/downward), symmetry (symmetrical/asymmetrical move adjustments), modality (unimodal/multimodal dimension adjustments), and duration (short term/long term) [23].

Furthermore, communicators' intentions and motives to adjust their communication (psychological adjustments) are considered subjective (individuals' perception of behavioral shifts), and linguistics adjustments are considered as objectives (observable shifts in behavior). CAT also posits that communicators tend to partially or fully accommodate those they like, trust, respect, and perceive interlocutors' needs and characteristics [25]. Stereotyped expectations and incongruent communicative behaviors can result in over or under-accommodating each other due to over-adjusting or not adjusting sufficiently [23]. Five socio-linguistic strategies have been proposed by CAT to enact communication adjustments (accommodation-non-accommodation) [26]. Primarily, approximation entails modifying one's vocal and nonverbal behaviors to conform to (convergence) or depart from the partner's (divergence). Second, interpretability: adapting the conversation topic to the partner's cognized or articulated understanding. Interactants can adjust their level of knowledge to that of their partners by avoiding technical jargon, simplifying language, or speaking loudly to clarify the words under investigation. Third, interpersonal control: this strategy is demonstrated by the use of interruptions to remind the partner of their duties and responsibilities (role relations, relative power, and status). Fourth, discourse management: this adjustment method may be influenced by the partner's perceived conversational needs, such as topic selection that is mutually beneficial to all parties involved. Fifth, the emotional expression: this technique can be used to reassure, support, and console another individual's emotional needs [27, 28].

CAT has been applied to study the interaction between patients and providers. It has also been used to study the intergenerational contexts of people with disabilities and their accommodation strategies [28]. One study employed semi-structured narrative interviews with 12 international interns in a hospital in Ohio. The study applied grounded theory to uncover how doctors used convergence to bridge intercultural and intergroup divides (e.g., repeating information, altering the manner of speech, or using non-verbal communication cues) [29]. Another study found that nurses used basic terminology and slowed down interactions with patients who were reluctant to communicate. This may frustrate patients who understand but cannot answer due to handicaps [30]. According to the best of our knowledge, CAT has never been used in the maternity ward to study the communication between healthcare providers and patients during childbirth. As a result, this study is a first-of-its-kind investigation on the accommodating behavior of healthcare providers in Bangladesh when communicating with patients in the maternity ward.

## Methods

### Study setting

Bangladesh's health system is pluralistic (with four providers), where the public (government-funded) and private sectors dominate. Healthcare is provided by non-profit private sector organizations and international development organizations in certain designated locations.

Both the public and private sectors have functional sub-divisions to accommodate their various infrastructures at the administrative district level. This study, however, considered only public sector health providers who provide maternal and healthcare services at various levels. The public sector offers maternity and pediatric care through community clinics that refer serious patients to union or upazila health complexes. Government-funded upazila (sub-district) health complexes (UHC) provide crucial maternity healthcare under the authority of the district civil surgeon's office. They have an inpatient capacity of roughly 30–50 beds. Although healthcare facilities at this level are still insufficient, patients are directed to a higher level of care (District general hospitals) if their condition is complicated [31]. The study collected data from healthcare providers that provide critical services to the maternity and child healthcare unit at one of the upazila health complexes and one of the district general hospitals in the Chattogram division of Bangladesh. A district general health hospital (DGH) is a secondary health facility that provides reproductive healthcare services such as gynecological and emergency obstetric care. Upazila health complexes (UHCs) are considered as the primary service point where professional healthcare providers provide facility-based deliveries with government-inflicted nominal user fees and receive preventive care and counseling [31].

## Study design and selection of study participants

This explorative study adopted an in-depth interview technique using a semi-structured interview guide. The study used a convenience sampling technique and recruited healthcare providers who were directly involved with patients and their relatives in the maternity ward in Bangladesh. The study interviewed doctors, midwives, and nurses who had a significant role in maternal health service operations. The study did not include health providers not associated with delivering maternity care. The study involved 10 participants (6 participants from UHC and four from DGH; two medical doctors, three midwives, and five senior nurses, all are women) who had different roles in the maternity ward. The number of participants was determined based on the researcher's resources, although the researcher does not claim that this study has reached any saturation threshold. The study encountered saturation to some extent; however, inflicting saturation might be a little far-fetched since a limited number of participants were available for the study. Initially, it was targeted to include more participants; however, due to covid-19, it was difficult to reach many as health providers have the additional duty of handling other patients. Participants from one UHC and DGH were studied and selected based on convenience; therefore, some response uniformity was predicted. The choice of these health facilities was intended to have a richer understanding of the study phenomenon in different levels of public health facilities rather than a comparison of both facilities. The study emphasized providers who have extensive interactions with patients during in-service operations, such as nurses/midwives who give services at night or when crowding or patient overload make service delivery problematic. The selection of participants was guided by the UHC head and the DGH resident medical officer. The study purposefully included a varied set of participants (doctors, nurses, and midwives) in order to explore and understand the differences in perceptions of patient communication and their corresponding accommodation techniques.

## Ethical approval

The study received ethical approval from the ethical committee of Hong Kong Baptist University, Hong Kong, and Noakhali Science & Technology University, Bangladesh. Participants' verbal and written consent was taken accordingly after formally discussing the study objectives, data anonymization, and use of pseudonyms. For keeping participants' anonymity

and personalized information confidential, the names of the two health facilities were not disclosed.

## Data collection process and analysis

Following participant selection with the assistance of hospital administration, the principal researcher (PR) and his graduate research associate (RA) contacted participants to schedule an interview. The interviews took place on hospital premises, preferably in a room free of visitors provided by the hospital administration. The interview took approximately 30–50 minutes. The participants were given sufficient time and ample opportunity to express themselves freely, and the interviews were conducted in their native language (Bengali). Throughout the interview process, follow-up questions (probing) were asked to elicit more information or clarification in order to eliminate ambiguity. The interviews with UHC participants were audio recorded with the participants' written and verbal agreement, and afterward, responses were transcribed verbatim. However, because participants from DGH did not agree to record the interview, field notes of the responses were collected during the interview. An interview guide based on current literature on patient-provider communication aided the interview (S1 File). The interviews were performed in the local language, and the order of questions varied depending on the conversation's natural flow. The interviews focused on the providers' perceptions of the engagement, possible facilitators and barriers to effective communication, and the type of accommodation methods they used to connect with patients. The interview guide included questions such as:

- What are the possible facilitators and barriers to effective communication with patients?

- How important do you think to comfort patients emotionally and share the medical decision to facilitate decision-making?

The PR performed all the interviews in person and kept the digital records and field notes for further processing. The PR conducted a pilot interview to ensure the questions were straightforward and did not cause concern. Participants were given random numbers to ensure confidentiality. MAXQDA 2020 qualitative data analysis software was used to manage the written transcripts. The PR did all the transcriptions and checked multiple times to avoid any loss of important data. PR generated tentative labels (open coding) to understand the essence of the interview responses (commonalities) and then generated preliminary codes (identifying key phrases) and sub-categories (axial coding). The graduate research associate (RA), experienced in qualitative studies, assisted the PR in the whole study and data analysis process. The RA also codes the dataset independently after verifying the transcripts; thus, any inconsistency or incongruity in the coding process is reconciled and addressed after thorough discussion. It helped validate the transcription process from multiple angles, focus on the interview narratives, and draw conclusions. Any discrepancy was extensively reviewed and addressed after reviewing the transcripts, records, and detailed field notes. After finalizing the codes and categories, inductive themes were generated by PR by applying Braun and Clarke's thematic analysis approach [32]. An excerpt of the thematic analysis process is presented briefly in Table 1.

## Results

### Inadequate healthcare facilities

When questioned about the elements that influence the level and pattern of communication with patients, most participants said that limited medical services and health facilities were

**Table 1. Thematic analysis.**

| Process of thematic analysis | | |
|---|---|---|
| Themes | Sub-categories | Illustrative quotes |
| Inadequate healthcare facilities | • Insufficient hospital beds and shortage of workforce<br>• Heavy workload<br>• Medical service limitations (e.g., lack of diagnostic facilities, drugs, etc.)<br>• Unimodal convergence<br>• Non-accommodation | *"When patients arrive and discover that there are no open beds and limited facilities, they obviously become frustrated. We attempt to convey to them our helplessness. Plus, the hospital does not provide all kinds of drugs and lacks diagnostic test facilities."*<br>• One senior nurse demonstrated limited service when asked about the start of communication process with patients. |
| Poor attitude and behavioral response | • Demeaning and disrespectful attitude<br>• Feel threatened and neglected<br>• Lack of health awareness and understanding of a medical situation<br>• Dispute divergence<br>• Asymmetrical adjustments | *"Sometimes, we do not receive the respect we deserve. Sometimes, I feel we are not worthy of respect and can be taken for granted."*<br>• One senior midwife expressed concern when asked about how some patients and their attendees behave. |
| Psychological accommodation | • Emotional convergence<br>• Supportive care and psychological adjustments<br>• Interpretative strategies<br>• Verbal and non-verbal cues<br>• Discourse management | *"It's our duty to provide emotional support to our patients. Of course, it's a difficult time for them, and we try to make them calm and talk it out. We use hand gestures, eye contact, or even touch hands to provide them comfort."*<br>• One gynecologist exemplified when asked about how they provide emotional support. |

major factors. Patients and clinicians interact from the moment they enter the unit via the emergency desk. However, with only 50 beds at UHC and 250 at DGH, there are times when the hospital cannot accommodate every patient. Patients may be irritated, dissatisfied, and deprived if compelled to get service on the hospital floor. Three senior nursing supervisors at DGH affirmed that there are always more patients than beds available. Most of the time, the patient is accompanied by many attendees who do not want to leave the hospital premises (Table 1). When asked about how they adjust or accommodate patients and their attendees in those times, one nursing supervisor who has working experience of more than 15 years in different parts of the country replied that:

"We try to diverge away from that topic and remain silent not to make the situation go out of hand. We are helpless in that regard. What's more frustrating is that (attendees) do not want to leave the maternity ward and make everything difficult by not maintaining visiting hours. On top of that, they repeatedly ask questions that we sometimes ignore to answer. We are human, too. Managing conversation is a big task for us; we have to remain vigilant and cautious so that we do not fall into any trap (conversational misunderstanding). Besides, there are no security guards that make the compound insecure."

The lack of ward boys, cleaners, sweepers, and aya (female workers) in UHC has made maintaining cleanliness in the ward and hospital difficult. Therefore, the UHC ward mostly remains crowded, unclean, and messy at times. The patients' limited privacy is another barrier to positive interaction. Most patients do not get any private cabin and use a common bathroom, which makes them feel disgusted. Moreover, UHC lacks modern cesarean section instruments and setups, and crucial empty positions like surgery or medical consultant do not help them either. Patient care requires teamwork and 24-hour logistic assistance, which is not provided at the ground level in health complexes; hence many cases are referred to the district

level institution, expressed by the only junior consultant (gynecologist) available at the UHC. UHC has limited medical supplies (drugs and equipment) and only provides primary and preventive care services. So, they refer critical or emergency patients to higher facilities, generally well-equipped district hospitals. To explain this matter, one midwife expressed who had vast working experience of more than a decade expressed frustration this way:

> "Another issue is that we don't have all sorts of drugs, so when we ask people to bring their own (from outside pharmacies), they claim it's a government hospital, and they should get all kinds of medicines. We only provided them with the available medicine in the hospital. The health centers are not always equipped with testing facilities as well. So, some diagnostic tests are also needed to be done in private diagnostic centers."

In response to the adjustment process in the face of service limitations, she noted that they all tend to converge on the common constraints with patients and try to help patients think rationally (about the prevalent limitations). They agree with the patients about the systematic inadequacy of the service operation that they are aware of. According to CAT theory, this is a unimodal and symmetric adjustment that may be reciprocated with patients; however, providers diverge (non-accommodation) when it comes to services they claim to provide adequately and appropriately. The divergence happens when patients or attendees claim that providers do not perform their duties accordingly.

## Poor attitude and behavioral response

In the context of limited-service facilities, some providers cited a negative attitude from patients as a hindrance to effective communication. Many times, patients attendees try to exert influence on doctors to secure additional advantages that providers are unable or unwilling to deliver. This is what a senior midwife, who has worked in various hospitals for over two decades, had to say:

> "Patients' relatives sometimes shout, and we don't get proper respect at times. They attempt to attack physically sometimes, in addition to verbal abuse. We do not have much security. If we see some patients who might get us into trouble, we try to handle them calmly and talk it out so that the situation does not go out of control. They claim that they do not get any service coming here. They call us "butchers." They try to intimidate us by calling powerful people. They act like, "call this person, call that person to scare us,"; but we have already given them the required service, and they fail to understand that. Sometimes we feel that we don't even have any voice."

The fact that so many women arrive at the hospital so late in their pregnancies has also been a source of frustration for healthcare providers. It could be due to poor access to resources or lack of awareness. People from lower socioeconomic backgrounds are more likely to use public facilities than those from wealthier social backgrounds. There are occasions when people lack the appropriate information and awareness when it comes to medical concerns. It can be challenging for service providers to connect effectively. To deal with them, providers used an approximation method (changing verbal and nonverbal behaviors) that may be tiring in a stressful setting. It was also important to note that nurses and midwives were the ones with whom the dispute mainly occurred; on the other hand, doctors rarely reported any disrespect or concerns regarding patients or their attendees' attitudes and behaviors. It could be due to nurses and midwives supervising maternity wards and frequently interacting with patients, or it could be owing to their rank or perceived professional

stature in the health service operation. Doctors, on the other hand, visit patients according to their allotted schedules. Patients and attendees may have the impression that nurses and midwives are on a par with or lower than their socioeconomic position, although they do not have this impression of doctors.

## Psychological accommodations

When questioned about their emotional support for patients, most healthcare providers claimed they demonstrate a positive attitude and attempt to be kind and compassionate. They educate patients and their caregivers about their rights and responsibilities and encourage them to seek medical attention when necessary. They also provide help for the underprivileged and inform them about the need to give birth at a health facility to prevent maternal morbidity and mortality. According to two doctors, women who are unsure if they should visit a health institution because of religious beliefs or cultural conventions may benefit from increased public awareness. The majority of health care personnel, excluding doctors, are fluent in the language of most patients. Patients' companions, particularly younger ones, were brought in to help with language barriers, especially those who could grasp and relay the information to the patients. One doctor in DGH replied that she often uses non-verbal communication (e.g., hand gestures, eye contact, demonstration, etc.) when she does not fully understand the patients' words. The interview responses regarding language issues indicate a convergent accommodation strategy where providers try to understand the patients' language and problems and adjust their own verbal and non-verbal behaviors. According to the situation, providers also used convergence strategies like repeating information where necessary and using divergence strategies to avoid troubles like not answering attendances' angry remarks on the hospital's systematic issue that they do not have control over. Providers tend to interpret the overall situation and try to make them understand; however, there were times when it became difficult to convey the message as attendances made life difficult by shouting that they had been ignored. The only gynecologist present in the UHC noted that:

> "If we suspect anything untoward might happen, we try to explain everything to them. They usually get our message about the support we want to provide. We strive to reassure and console them."

From this response, it is observable that she had applied interpretability strategies of communication accommodation theory. She also replied that it becomes easier to fully accommodate those who fully support their effort and the services they receive. It was also evident that the providers used an emotional expression strategy since almost all providers stated how they used to provide emotional support, comfort, and assurance to their patients. The doctors said that they are completely aware of the services they offer and that most patients conduct themselves appropriately. And when a disagreement arises, the hospital administration works to involve both parties and settle the situation so that everyone feels accountable. However, while that sounds excellent on paper, it appears clunky and noisy in practice at times.

## Discussion and conclusions

### Discussion

The study investigated the level of communication between healthcare providers and patients in two health facilities that provide primary and secondary care, respectively. The study looked into how the communication accommodation theory and strategies can be used to explain these interactions. The codes highlight key themes such as health service and facility

limitations, patients' attitudes towards providers, and providers' accommodation strategies to comfort patients' emotional and psychological needs.

Most providers reported health service facilities and treatment-related constraints, especially in UHCs. UHC's lack sophisticated medical equipment (primarily provide outpatient primary level care), and some important medical positions are vacant, which further limits the service optimization. In order to avoid miscommunication with patients, providers indicated that accommodation begins with helping patients understand the available services they provide to achieve unimodal, symmetric, and beneficial adjustment. For instance, there was no in-house anesthesiologist and medical surgeon present in UHC to conduct any surgery related to childbirth, so letting patients know about it would surely help patients understand. One study on nurses' attitudes toward patients in a rural district of South Africa found that some nurses disliked nursing due to staff shortages, high patient loads, absenteeism, and a lack of interpersonal communication [33]. These service constraints force patients to go to district-level health facilities, which patients do not want to go to for various reasons. Patients' non-reliance on accepting the limitations encourages providers to diverge away and converge to a unimodal direction when providers get to comply with patients' grudge. The shortage of healthcare workers grows increasingly pervasive and visible during this pandemic. Bangladesh is no exception since the country was already in a state of distress in terms of providing quality care to its citizens [34]. Moreover, some nurses and midwives report feeling under pressure in certain situations due to the power dynamics in patient-provider interactions. Doctors are well-trained and educated compared to other staff, and people respect them. It also could be an indication that people are less appreciative of nurses and midwives than they are of doctors. It results in downward and asymmetrical accommodation from some nurses and midwives who think they do not get deserved respect from the patients and their attendees. There were seen to have trust issues between the patients and the nurses/midwives, which were not felt from the doctor's perspective.

In the service limitation aspect, some providers converge univocally (symmetry) to accept the fact that there are visible shortages that healthcare struggles to cover. Although DGH providers claim to provide most treatments and drugs, Islam & Biswas (2014) have found a pattern of systematic failing to cover all medical charges. Government spending contributes to almost one-third of total health costs. The patients have to bear the remaining out-of-pocket expenses, which makes them cost-concerned, resulting in not visiting health facilities [35]. Patients also resent paying for medicines and diagnostics in private institutions, which they regard costly. Several nurses and midwives expressed displeasing experiences of possible verbal "face-off" with attendees over the hospital beds, cleanliness, out-of-pocket expenses, diagnostic testing facilities, etc. They employ divergence strategy mostly while that happens when they fail to convince the patients and their raging attendees after attempting to make them understand. Patients sometimes verbally argue with nurses and midwives about their care; this could be related to the nurses'/midwives' long hours with inpatients and their perceived poor professional status.

It was interesting to analyze providers' accommodation strategies amidst all that was happening around. Health providers argued that they attempt to communicate with patience, care, and affection and emphasize counseling, health behavior improvement, and verbal or non-verbal emotional support [36]. Providers alter their verbal and nonverbal responses in order to corroborate or disassociate themselves from certain situations suggestive of implying CAT's approximation method. Provider's responses to accommodate patients diverge away on many occasions, as discussed above when pacifying patients or responding to criticism. Providers reported that they mostly remain silent and try not to focus on that. This indication of divergence can assist in clarifying the dialogue and highlighting distinctions that can assist the

interactants in comprehending one another in some instances [25]. Although constraints such as conversational norms, difficulty understanding acronyms, technical jargon, and disclosure of medical information are common in patient-provider contact [29], providers indicated that they handle situations effectively. Interpretability strategies (unimodal, short-term convergence) seem to be used by providers when they need to explain medical situations to their patients. As accommodation is desirable in patient-provider communication [36], the providers acknowledged that they are unable to talk to many people to entertain their repeated attempts to understand any issue, making it a possible non-accommodation and divergence. One doctor also expressed that the patient's decision-making inability also hampers communication.

The study only considered providers' viewpoints, which could be a limitation as patients' reflections might have added value. The relationship dynamic between patients and nurses/midwives is interesting, and patients' perception and experience of interaction might invigorate the understanding of the interaction process. The study only looked at two health complexes with a limited number of participants; thus, its findings must be used carefully. Also, the author does not suggest that effective patient-provider communication can address all problems in healthcare; rather, it can be a catalyst for beneficial connection. Further research involving more participants may facilitate more insight into this very important research. The study tried to understand the accommodation patterns of health providers based on the CAT theory's theoretical explanations, which do not imply that the health facilities adopted the CAT approach to their patients in real.

## Conclusions

Provider's perception of interaction with patients and their adopted accommodation strategies for providing healthcare service delivery were studied and analyzed using communication accommodation strategy. In response to systematic limitations of health facilities and heavy workload, the providers, especially nurses and midwives, reported poor attitudes and neglect from some patients. The patients' attitudes towards providers also indicated a power dynamic in the service operation where midwives and nurses ingress into the lower category. Health providers endorsed different approaches to accommodate patients' emotional and psychological needs and interpret medical situations through verbal and non-verbal cues. However, it's important to address all the important factors that constrain the interaction to facilitate better communication. Prompt delivery of health services and awareness initiatives to urge patients and visitors to cooperate with medical procedures can be useful. There would be a perfect accommodation in an ideal world where patients receive the care they desire, and providers offer conscientious service as a repercussion of patients' desires; Bangladesh should endeavor to attain that.

## Policy and communication implications

This study expressed healthcare practitioners' perspectives on interpersonal communication with patients and their relatives, a group that is understudied and difficult to reach. In response to several studies on patients' discontent with healthcare, especially in a resource-constrained setting, the provider's perspective is vital [3, 8, 16, 33]. The study revealed that physicians accommodate patients, to a certain extent, which may be useful in targeting and improving patient-provider interaction, a key feature of health service quality. These study findings also warrant the importance of effective patient management by the providers. Efforts are needed to boost the nursing staff's perceived professional status in Bangladesh, as well as to manage their professional capacity development. Patient management, respectful

care, and communication skill training can help health providers accommodate patients according to the defined set of standards. Also, health facilities can initiate a patient awareness program that also targets attendees so that they collaborate with the current settings as much as possible. The healthcare sector can also consider facilitating providers' workload and maintaining a power balance in service management.

## Supporting information

**S1 File. Interview guide.**
(DOCX)

## Acknowledgments

The author would like to acknowledge the support of the district administration, local administration, upazila health complex, district general hospital authorities, and participants who willingly participated in this study. The author is indebted to the upazila health & family planning officer (UHFPO) and the resident medical officer (RMO) of the health facilities who helped facilitate the whole study process. The author would like to acknowledge Dr. Dominic Yeo of Hong Kong Baptist University for his valuable suggestions and comments on this work.

## Author Contributions

**Conceptualization:** Md. Ruhul Kabir, Kara Chan.

**Data curation:** Md. Ruhul Kabir.

**Formal analysis:** Md. Ruhul Kabir.

**Funding acquisition:** Md. Ruhul Kabir.

**Investigation:** Md. Ruhul Kabir.

**Methodology:** Md. Ruhul Kabir, Kara Chan.

**Project administration:** Md. Ruhul Kabir.

**Resources:** Md. Ruhul Kabir.

**Software:** Md. Ruhul Kabir.

**Supervision:** Md. Ruhul Kabir.

**Validation:** Md. Ruhul Kabir, Kara Chan.

**Visualization:** Md. Ruhul Kabir.

**Writing – original draft:** Md. Ruhul Kabir, Kara Chan.

**Writing – review & editing:** Md. Ruhul Kabir, Kara Chan.

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
