## [Decision Letter · Decision Letter 0]

31 Jan 2022

PONE-D-21-37993“Do We Even Have a Voice?” Health providers’ perspective on the patient accommodation strategies in BangladeshPLOS ONE

Dear Dr. Ruhul Kabir,

Thank you for submitting your manuscript to PLOS ONE. After careful consideration, we feel that it has merit but does not fully meet PLOS ONE’s publication criteria as it currently stands. Therefore, we invite you to submit a revised version of the manuscript that addresses the points raised during the review process.

We look forward to receiving your revised manuscript.

Kind regards,

Paavani Atluri

Academic Editor

PLOS ONE

Journal Requirements:

2. During our internal checks, the in-house editorial staff noted that you conducted research or obtained samples in another country. Wile we appreciate that in the Ethics statement you have indicated that "The study was also approved by the review board of the two health facilities where the study was conducted".  Please check the relevant national regulations and laws applying to foreign researchers and state whether you obtained the required permits and approvals. And please revise your statement  to include the full name of the ethics committee/institutional review board(s) that approved your specific study and to confirm that your named institutional review board or ethics committee specifically approved this study.

In addition, please ensure that you have suitably acknowledged the contributions of any local collaborators involved in this work in your authorship list and/or Acknowledgements. Authorship criteria is based on the International Committee of Medical Journal Editors (ICMJE) Uniform Requirements for Manuscripts Submitted to Biomedical Journals - for further information please see here: https://journals.plos.org/plosone/s/authorship.

Reviewers' comments:

Reviewer's Responses to Questions

**Comments to the Author**

1. Is the manuscript technically sound, and do the data support the conclusions?

Reviewer #1: Yes

Reviewer #2: Yes

2. Has the statistical analysis been performed appropriately and rigorously? 

Reviewer #1: N/A

Reviewer #2: I Don't Know

3. Have the authors made all data underlying the findings in their manuscript fully available?

Reviewer #1: No

Reviewer #2: Yes

4. Is the manuscript presented in an intelligible fashion and written in standard English?

Reviewer #1: Yes

Reviewer #2: Yes

5. Review Comments to the Author

Reviewer #1: Overall an excellent manuscript. However, few suggestions or comments:

1. Discussion should be based on CAT, although results section does include integration of CAT.

2. In the methodology section, how and on what grounds the participants were selected? What inclusion and exclusion criteria

was followed?

3. If no saturation was achieved, then justify 10 participants for the study?

4. Selection of hospitals was done on what grounds? Are both hospitals similar? From the methodology section it appears that

1 is secondary health institution and 1 is primary health institution. How can you justify using 2 different levels of health

institutions? For study to have some uniformity, both institutions have to be of similar nature.

5. How reliability and validity of the data (transcriptions) was ensured.

6. The study has one perspective only that is service provider perspective. In my opinion if patients are also included the

study would have had more strength. This is one future direction that needs to be explicitly stated in future directions.

Reviewer #2: The manuscript talks about the experience of mistreatment, discrimination, and disrespect in a resource constrained setting in a developing country. This manuscript concentrates on the degree of interaction with pregnant women particularly in the context of accommodating patients and their relatives to improve communication between the parties of health care as well as patients. Manuscript also mentions of systematic limitations of health care facilities and heavy workload reported by staff. Poor attitudes and neglect from some patients and providers perception of interaction with patients therefore their adopted accommodation strategies for providing health care services were studied. This was analyzed using communication accommodation strategy (CAT). Communication accommodation theory which is one of the prominent behavioral theories of communication and the author provides a thematic analysis in table 1. Provider’s perception of patient interaction and their adopted accommodation strategies were studied. Due to systematic limitations of health care facilities and heavy workload especially nurses and midwives reported poor attitudes and neglect from some patients. The health care providers also use different approaches to accommodate patients emotional and psychological needs through verbal and nonverbal clues. The author request for prompt delivery of health care services and awareness programs to urge patients and visitors to cooperate with medical procedures, so they can collaborate with current settings as much as possible. Health care sector can also consider facilitating providers workload and maintaining a power balance and service management.

6. PLOS authors have the option to publish the peer review history of their article (what does this mean?). If published, this will include your full peer review and any attached files.

Reviewer #1: No

Reviewer #2: No

---

## [Author Response · Author response to Decision Letter 0]

11 Feb 2022

The authors would like to thank the Editor and the two reviewers for their valuable comments and insights that helped to improve the manuscript. We have tried to answer all the issues mentioned by the reviewers and update some parts based on the suggestions received. Please find the details below where we have tried to clarify points one by one. Two references added to the (Introduction part, Reference number: 11, 13) reference list due to their relevance to the literature. 

Comments from reviewers and responses from the author:

1. Discussion should be based on CAT, although results section does include integration of CAT.

Response: Thank you for this important comment. We have updated the discussion section, which focused more on CAT, although CAT was discussed in the middle part of the discussion section already.

2. In the methodology section, how and on what grounds the participants were selected? What inclusion and exclusion criteria was followed?

Response: The study used a convenience sampling technique and considered healthcare providers who were directly involved with patients and their relatives in the maternity ward in providing delivery care or other care related to maternal and childcare. The study interviewed doctors, midwives, and nurses who had a significant role in maternal health service operations. This study did not consider health providers who were not directly involved with maternity care. 

3. If no saturation was achieved, then justify 10 participants for the study?

Response: Initially, it was planned to include more participants for this study. But due to covid-19, it was difficult to reach health providers even if they work in maternity care mostly. They were involved in other patients' care as well, which made them a very difficult group of people to reach in this unprecedented period of time. The study did encounter saturation to some extent; however, inflicting saturation might be a little far-fetched since only limited participants were available. The final number of participants was decided based on their availability, timing, and free schedule. No generalization of the findings was targeted. The study tried to deliberately include a diverse group of participants (doctors, nurses, and midwives) to explore and understand the diversity in perceptions of communication with patients and their diverse accommodation strategies.

4. Selection of hospitals was done on what grounds? Are both hospitals similar? From the methodology section it appears that 1 is secondary health institution and 1 is primary health institution. How can you justify using 2 different levels of health institutions? For study to have some uniformity, both institutions have to be of similar nature.

Response: The study targeted public health facilities in Bangladesh, so in that aspect, both the facilities were publicly funded and selected based on convenience of access. The study was not intended to compare the primary and secondary level of care; it was intended to assess health providers' perceptions about their interaction with patients in different levels of healthcare and how they accommodate their patients based on their level of facilities. Of course, similarities-dissimilarities were discussed, but the basis was to have a deeper understanding of common grounds.

5. How reliability and validity of the data (transcriptions) was ensured.

Response: Probing questions were asked to participants to prevent ambiguity or double meaning. Data curation and analysis were assisted by one research associate who also independently read and coded the data set to validate the process through multiple angles. We focused only on interview responses to limit researcher’s subjectivity. The principal researcher and associate went through all the data sets to find out any incongruity in their coding based on the initial codebook developed by the principal researcher. The process was rechecked multiple times (from field notes to transcriptions to theme development) till the generation of themes, so that discrepancies could be reconciled and resolved. The process is thoroughly discussed in the data analysis section.

6. The study has one perspective only that is service provider perspective. In my opinion if patients are also included the study would have had more strength. This is one future direction that needs to be explicitly stated in future directions.

Response: I agree with you completely and therefore, I already included it in the limitation section (last part of the discussion). Conclusion and policy implication is based on the current study findings. A future research is under consideration which will also include patients’ perspectives. Thank you for this great suggestion.

7. The manuscript talks about the experience of mistreatment, discrimination, and disrespect in a resource constrained setting in a developing country. This manuscript concentrates on the degree of interaction with pregnant women particularly in the context of accommodating patients and their relatives to improve communication between the parties of health care as well as patients. Manuscript also mentions of systematic limitations of health care facilities and heavy workload reported by staff. Poor attitudes and neglect from some patients and providers perception of interaction with patients therefore their adopted accommodation strategies for providing health care services were studied. This was analyzed using communication accommodation strategy (CAT). Communication accommodation theory which is one of the prominent behavioral theories of communication and the author provides a thematic analysis in table 1. Provider’s perception of patient interaction and their adopted accommodation strategies were studied. Due to systematic limitations of health care facilities and heavy workload especially nurses and midwives reported poor attitudes and neglect from some patients. The health care providers also use different approaches to accommodate patients emotional and psychological needs through verbal and nonverbal clues. The author request for prompt delivery of health care services and awareness programs to urge patients and visitors to cooperate with medical procedures, so they can collaborate with current settings as much as possible. Health care sector can also consider facilitating providers workload and maintaining a power balance and service management.

Response: Thank you for your comment.

---

## [Decision Letter · Decision Letter 1]

15 Jun 2022

PONE-D-21-37993R1“Do We Even Have a Voice?” Health providers’ perspective on the patient accommodation strategies in BangladeshPLOS ONE

Dear Dr. Kabir,

Thank you for submitting your manuscript to PLOS ONE. After careful consideration, we feel that it has merit but does not fully meet PLOS ONE’s publication criteria as it currently stands. Therefore, we invite you to submit a revised version of the manuscript that addresses the points raised during the review process.

We look forward to receiving your revised manuscript.

Kind regards,

Paavani Atluri

Academic Editor

PLOS ONE

Journal Requirements:

Reviewers' comments:

Reviewer's Responses to Questions

**Comments to the Author**

1. If the authors have adequately addressed your comments raised in a previous round of review and you feel that this manuscript is now acceptable for publication, you may indicate that here to bypass the “Comments to the Author” section, enter your conflict of interest statement in the “Confidential to Editor” section, and submit your "Accept" recommendation.

Reviewer #3: (No Response)

Reviewer #4: All comments have been addressed

2. Is the manuscript technically sound, and do the data support the conclusions?

Reviewer #3: Yes

Reviewer #4: Partly

3. Has the statistical analysis been performed appropriately and rigorously? 

Reviewer #3: Yes

Reviewer #4: N/A

4. Have the authors made all data underlying the findings in their manuscript fully available?

Reviewer #3: Yes

Reviewer #4: Yes

5. Is the manuscript presented in an intelligible fashion and written in standard English?

Reviewer #3: Yes

Reviewer #4: Yes

6. Review Comments to the Author

Reviewer #3: The manuscript can be improved further if the comments inserted in the document are addressed/considered.

Reviewer #4: The authors addressed the points raised by the reviewers. However the limitations imposed by the small and not so diverse group of interviewees in the study persist in some extent.

I also suggest a final review of the text. There are some minor mistakes which should be corrected.

7. PLOS authors have the option to publish the peer review history of their article (what does this mean?). If published, this will include your full peer review and any attached files.

Reviewer #3: **Yes: **Manoja Kumar Das

Reviewer #4: No

---

## [Author Response · Author response to Decision Letter 1]

16 Jun 2022

Reviewer's comments:

Reviewer #3: The manuscript can be improved further if the comments inserted in the document are addressed/considered.

Reviewer #4: The authors addressed the points raised by the reviewers. However the limitations imposed by the small and not so diverse group of interviewees in the study persist in some extent.

I also suggest a final review of the text. There are some minor mistakes which should be corrected.

Author’s response: Thank you for your comments and suggestions which we belief have made us more focused to improve our work further.

We addressed all the comments/suggestions made by the reviewers and tried to improve the manuscript. We went through the manuscript again, according to your suggestion, to find out and correct minor mistakes, where applicable. We agree with you regarding the relatively small sample size, to some extent. We tried to make the sample as heterogenous as possible involving doctors, nurses, and midwives as participants within our limits.

---

## [Editor Report · Decision Letter 2]

8 Jul 2022

“Do We Even Have a Voice?” Health providers’ perspective on the patient accommodation strategies in Bangladesh

PONE-D-21-37993R2

Dear Dr. Kabir,

We’re pleased to inform you that your manuscript has been judged scientifically suitable for publication and will be formally accepted for publication once it meets all outstanding technical requirements.

Kind regards,

Paavani Atluri

Academic Editor

PLOS ONE
---

## [Editor Report · Acceptance letter]

22 Jul 2022

PONE-D-21-37993R2 

“Do We Even Have A Voice?” Health Providers’ Perspective On The Patient Accommodation Strategies In Bangladesh 

Dear Dr. Kabir:

I'm pleased to inform you that your manuscript has been deemed suitable for publication in PLOS ONE. Congratulations! Your manuscript is now with our production department. 

Kind regards, 

on behalf of

Dr. Paavani Atluri 

Academic Editor

PLOS ONE